# Conservation aquaculture as a tool for imperiled marine species: Evaluation of opportunities and risks for Olympia oysters, *Ostrea lurida*

**April D. Ridlon**[1]*, **Kerstin Wasson**[2,3], **Tiffany Waters**[4], **John Adams**[5], **Jamie Donatuto**[6], **Gary Fleener**[7], **Halley Froehlich**[8], **Rhona Govender**[9], **Aaron Kornbluth**[10], **Julio Lorda**[11,12], **Betsy Peabody**[13], **Gifford Pinchot IV**[14], **Steven S. Rumrill**[15], **Elizabeth Tobin**[16], **Chela J. Zabin**[17], **Danielle Zacherl**[18], **Edwin D. Grosholz**[19]

1 Science for Nature and People Partnership and National Center for Ecological Analysis and Synthesis, University of California Santa Barbara, Santa Barbara, California, United States of America, 2 Elkhorn Slough National Estuarine Research Reserve, Watsonville, California, United States of America, 3 Ecology and Evolutionary Biology University of California—Santa Cruz, Santa Cruz, California, United States of America, 4 Global Aquaculture, The Nature Conservancy, Arlington, Virginia, United States of America, 5 Sound Fresh Clams and Oysters, Shelton, Washington, United States of America, 6 Community Environmental Health Program, Swinomish Indian Tribal Community, LaConner, Washington, United States of America, 7 Research and Development, Hog Island Oyster Co., Marshall, California, United States of America, 8 Ecology, Evolution & Marine Biology and Environmental Studies, University of California Santa Barbara, Santa Barbara, California, United States of America, 9 Species at Risk Program, Fisheries and Oceans Canada, British Columbia, Canada, 10 Officer, The Pew Charitable Trusts, Washington D.C., United States of America, 11 Facultad de Ciencias, Universidad Autónoma de Baja California, Mexicali, Mexico, 12 Tijuana River National Estuarine Research Reserve, Imperial Beach, California, United States of America, 13 Puget Sound Restoration Fund, Bainbridge Island, Washington, United States of America, 14 Chelsea Farms, Olympia, Washington, United States of America, 15 Marine Resources Program, Oregon Department of Fish and Wildlife, Newport, Oregon, United States of America, 16 Natural Resources Department, Jamestown S'Klallam Tribe, Sequim, Washington, United States of America, 17 Marine Invasions Research, Smithsonian Environmental Research Center, Belvedere Tiburon, California, United States of America, 18 Department of Biological Science, California State University Fullerton, Fullerton, California, United States of America, 19 Department of Environmental Science and Policy, University of California—Davis, Davis, California, United States of America

* ctenophores@gmail.com

**Data Availability Statement:** All relevant data are within the paper and its Supporting Information files.

## Abstract

Conservation aquaculture is becoming an important tool to support the recovery of declining marine species and meet human needs. However, this tool comes with risks as well as rewards, which must be assessed to guide aquaculture activities and recovery efforts. Olympia oysters (*Ostrea lurida*) provide key ecosystem functions and services along the west coast of North America, but populations have declined to the point of local extinction in some estuaries. Here, we present a species-level, range-wide approach to strategically planning the use of aquaculture to promote recovery of Olympia oysters. We identified 12 benefits of culturing Olympia oysters, including identifying climate-resilient phenotypes that add diversity to growers' portfolios. We also identified 11 key risks, including potential negative ecological and genetic consequences associated with the transfer of hatchery-raised oysters into wild populations. Informed by these trade-offs, we identified ten priority estuaries where aquaculture is most likely to benefit Olympia oyster recovery. The two highest

**Funding:** This work was funded by a David and Lucile Packard Foundation Grant (#2018-68222) to the Science for Nature and People Partnership. Additional support was provided for all co-authors from their respective organizations. The funders had no role in study design, data collection and analysis, decision to publish, or preparation of the manuscript.

**Competing interests:** While the following authors were affiliated with commercial companies - Gary Fleener, Hog Island Oyster Company; Gifford Pinchot IV, Chelsea Farms; and John Adams, Sound Fresh Clams and Oysters – they did not receive any salary support to contribute to the manuscript. This commercial affiliation does not alter their adherence to PLOS ONE policies on sharing data and materials.

scoring estuaries have isolated populations with extreme recruitment limitation—issues that can be addressed via aquaculture if hatchery capacity is expanded in priority areas. By integrating social criteria, we evaluated which project types would likely meet the goals of local stakeholders in each estuary. Community restoration was most broadly suited to the priority areas, with limited commercial aquaculture and no current community harvest of the species, although this is a future stakeholder goal. The framework we developed to evaluate aquaculture as a tool to support species recovery is transferable to other systems and species globally; we provide a guide to prioritizing local knowledge and developing recommendations for implementation by using transparent criteria. Our collaborative process engaging diverse stakeholders including managers, scientists, Indigenous Tribal representatives, and shellfish growers can be used elsewhere to seek win-win opportunities to expand conservation aquaculture where benefits are maximized for both people and imperiled species.

## Introduction

Marine biodiversity and the ecosystem services marine species provide are in decline globally, but it is not too late for these changes to be reversed [1]. Marine foundation species (e.g. kelp, mangroves, corals, oysters) are critical to the structure and resilience of coastal ecosystems, providing key ecosystem services to human communities around the world [2, 3]. Many marine foundation species have suffered severe population declines due to human activities including overfishing, habitat loss, and climate change (e.g. [4, 5]). Oysters act as foundation species by creating habitat for other estuarine species and providing key ecosystem services to human communities around the world [6–8]. In addition to ameliorating environmental stressors, such as dampening storm surges [9] and improving water quality via increased filtration [10], oysters have been a food source for people around the world for millennia, through both harvest of wild populations and aquaculture [11]. However, like many other marine foundation species, oyster populations have declined precipitously: in the United States, there has been an 88% loss in oyster biomass [12] and worldwide, an estimated 85% of oyster reefs have been lost, a figure exceeding the estimated loss of coral reefs [9]. Simultaneously, ever-rising global demand for protein, among other factors, is driving the rapid expansion of shellfish aquaculture [13–15], including oyster farming, while the restoration of native oyster species and wild oyster beds has become a priority for maintaining the health and ecosystem function of estuaries [16, 17].

The Olympia oyster (*Ostrea lurida*) is native to estuaries from British Columbia, Canada to Baja California, Mexico [18, 19], and a prime example of a foundation species whose populations have drastically declined. Olympia oysters historically created habitat for numerous estuarine and coastal species [20, 21], were harvested by Indigenous people [22, 23], and supported a vital fishery [24]. However, populations declined throughout its range due to over-harvesting and habitat degradation following European settlement [25, 26]. In some regions, populations have been reduced to 1% of their historic levels and face local extinction [10]. Thus, while Olympia oysters are still present in many estuaries, they no longer form dense, habitat-forming beds in many places. Wild populations of the native oyster are further challenged by the alteration of estuarine habitats, non-native predators, poor water quality, sedimentation, and lack of natural recruitment [27, 28]. Importantly, a recent synthesis revealed that recruitment limitation is the second biggest challenge to restoration success with Olympia oysters throughout their range, negatively affecting over 70% of all projects [28].

Conservation aquaculture–defined by Froehlich [29] as human cultivation of an aquatic organism for the planned management and protection of a natural resource–is a tool with enormous potential to benefit both nature and people, by simultaneously supporting populations of marine species and providing economic and social benefits to human communities. Conservation aquaculture emphasizes the importance of ecologically responsible methods to implement, and scientifically rigorous methods to evaluate, the use of aquaculture techniques that purposefully align with conservation goals. Its techniques also specifically seek to minimize the risks sometimes associated with conventional aquaculture. For example, conservation hatchery protocols address risks related to the release of hatchery-reared organisms, including preserving the genetic diversity of the wild population and minimizing the propagation of invasive species [30, 31]. Here, we focus specifically on the application of conservation aquaculture as a tool to aid the recovery of an imperiled species, although its definition and benefits can be much broader [29]. Conservation aquaculture programs have been successful in supporting the recovery of endangered fish (e.g. white sturgeon: [32]) and invertebrate populations (e.g. white abalone [33]), and aquaculture techniques have successfully supported restoration of declined oyster populations (e.g. eastern oyster *Crassostrea virginica* [34]; European flat oyster *Ostrea edulis* [35]). However, aquaculture has generally been used opportunistically, at sites where existing hatcheries or conservation organizations are located, without range-wide strategic planning to prioritize locations that would most benefit imperiled species.

In fact, conservation aquaculture is likely to be of greatest benefit in supporting imperiled populations where reproduction is a key bottleneck to population growth, and where conditions are favorable for survival of juveniles [36]. Aquaculture is only one of many conservation tools and may be most effectively used for species recovery when integrated into a holistic management strategy (e.g. alongside fishing regulations and/or habitat improvement planning; [37, 38]). Importantly, conservation aquaculture also comes with potential risks such as the loss of genetic adaptations of the wild population and modification of habitat by the placement of commercial gear or artificial reef structures [39]. These concerns were recently highlighted by efforts to use conservation aquaculture to restore populations of reef building corals [40]. Froehlich [37] emphasized the importance of exploring potential ecological trade-offs associated with conservation aquaculture and the need to collaboratively work with stakeholders to set shared conservation goals and priorities.

The use of aquaculture techniques to support the recovery of Olympia oyster populations is relatively limited to date. Efforts to produce oysters for restoration were pioneered in Oregon and Washington in the 1990s, through individual partnerships between managers and commercial or tribal hatcheries. The first conservation hatchery for this species, the Chew Center, was established in 2013, increasing capacity for restoration efforts in Washington via a partnership between the National Oceanic and Atmospheric Administration and Puget Sound Restoration Fund. A few aquaculture-assisted restoration projects have since been employed in central California and Oregon where estuary-wide recruitment failure is common [27, 36]. However, the majority of projects throughout the species' range have not utilized aquaculture to restore or enhance populations [28].

Likewise, since the onset of the cultivation of Olympia oysters, commercial aquaculture of the native oyster has remained very limited [24]. The Pacific oyster (*Crassostrea gigas*)—a hardy, fast-growing native of east Asia—has instead been the dominant commercially grown oyster on the West Coast [41]. While some commercial growers raise Olympia oysters as a specialty product, they may be more motivated by an interest in the ecology and heritage of the species than its current monetary value. However, consumer interest and demand for Olympia oysters is increasing, and with it the potential for additional growers to create a larger Olympia oyster market, with prices equal to or greater than Pacific oysters [42].

To our knowledge, there has been no thorough evaluation of the rewards versus risks of conservation aquaculture to guide strategic planning for any marine species on this coast. Here, we present a species-level, range-wide approach to strategically planning the expanded use of aquaculture as a tool to support recovery of Olympia oysters. Before funders, planners, regulators, conservation organizations, or growers move forward with aquaculture initiatives, there is a pressing need to conduct a robust, collaborative evaluation of whether and under what conditions the rewards outweigh the risks. Additionally, detailed spatial analyses are needed to identify the specific locations where conservation aquaculture efforts can be expected to have a high likelihood for success. A transparent and consistent evaluation process is essential to assess sites across the range of the species, so stakeholders can jointly identify those places where aquaculture is most likely to support species recovery and/or enhance local communities. Funders and conservation organizations require assistance to make strategic investments to gain the greatest impact on species recovery, resource management agencies want to know where risks are lowest, and growers want greater confidence in determining how and where their native oyster aquaculture investments can support conservation efforts. Decision-support tools have been developed to inform strategic planning for aquaculture in other regions [43, 44].

Our goal was to conduct strategic planning and develop a framework to evaluate the use of aquaculture to support recovery of Olympia oysters across the range of the species. We also assessed human dimensions of aquaculture, including the potential for community harvest and commercial production, within the framework of supporting recovery of wild populations. We designed an inclusive process of engaging stakeholders from the beginning, which is much more likely to yield results and reduce conflict [45]. We thus drew from and expanded upon the membership of the Native Olympia Oyster Collaborative (https://olympiaoysternet. ucdavis.edu/), which brings together local stakeholders united by the belief that a coast-wide perspective will lead to better conservation outcomes for the species. Our team of growers, tribal representatives, scientists, and conservation practitioners evaluated the potential of conservation aquaculture for this species, benefitting from both the local expertise represented by participants, and the commitment to coast-wide collaboration. Together, we explored the rewards and risks of aquaculture for this species, identified priority locations for new or expanded investment in this tool, and assessed human dimensions, to consider the types of aquaculture projects most appropriate for those locations. This collaborative approach, with jointly developed criteria applied consistently at a broad geographic scale, can serve as a model for other species where aquaculture may assist in recovery of wild populations.

## Methods and materials

### Collaborative process with stakeholder team

We engaged a diverse group of 30 stakeholders with expertise or interests related to Olympia oyster restoration, conservation, and production of native oysters for local food systems. Members of the stakeholder group were selected from the full geographic range of this species from British Columbia, Canada to Baja California, Mexico, based on their expertise in working with Olympia oysters, with attention to engaging stakeholders with different goals and perspectives. Most were already affiliated with the Native Olympia Oyster Collaborative and were known to our core team as a result of our earlier work establishing this network, though some were new partners referred to us by members. Our stakeholder group included conservation scientists, restoration practitioners, aquaculture specialists from non-profit organizations and state and federal agencies, marine resource managers, representatives from two Western Washington Treaty Tribes, and commercial oyster growers (S1 Table). To iteratively incorporate expertise

with ecological, social, and economic factors, we engaged the stakeholder group in three primary stages: (1) qualitatively identifying and refining the rewards and risks (i.e. trade-offs) for conservation aquaculture of this species, (2) selecting estuaries to evaluate, and (3) creating and scoring the conservation aquaculture indices for each estuary. We communicated through many small virtual meetings and through review of shared on-line documents, and also held discussions with the full group in a two-day virtual workshop. We used collaborative, iterative processes to gather the best available quantitative data, elicit the group's expert opinion, and/ or solicit the input of additional local stakeholders where appropriate through focused meetings in particular regions. Our work did not involve 'research on human subjects'; we did not collect information about participating members or study their individual perspectives, and thus did not require Institutional Review Board approval. Instead, the input from stakeholders with diverse expertise was synthesized as a collaborative activity with all stakeholders involved in the team effort.

### Rewards and risks of conservation aquaculture with Olympia oysters

Conservation aquaculture provides a suite of rewards and risks that are specific to the species and population under consideration. In March 2020, the stakeholder group met to explore the wide range of potential rewards and risks associated with conservation aquaculture of Olympia oysters in western North America. The range of potential rewards and risks was then condensed into broad categories based on the primary beneficiary: (1) Olympia oyster populations, (2) the coastal ecosystem, (3) Tribes / First Nations, and other local community members, (4) conservation practitioners, and (5) commercial growers. Due to the overlap of rewards and risks identified for Tribes / First Nations and local community members, these were later combined into one user group. This list was further condensed to focus on the rewards and risks that were specific to conservation aquaculture, rather than general rewards and risks that could also be applied to oyster restoration.

While the rewards and risks were often determined to have more than one recipient group, one primary recipient group was identified and listed. We separately present the rewards and risks relative to our focal beneficiary categories as they can differ according to which entities the project is designed to benefit. For example, projects including harvest or commercial production will have different associated rewards and risks than projects focused solely on restoration. Likewise, there are different rewards and risks involved if project goals include community engagement than if the project is conducted by staff at a conservation organization. Our stakeholders represented, and were interested in the full spectrum of project types and goals, and we evaluated the rewards and risks associated with each of them. From the identified rewards and risks, we then formed recommendations for minimizing risks and maximizing rewards.

### Estuary selection

Olympia oysters are almost entirely limited to estuaries; there are no large populations along the open coast [18, 24]. Consequently, we focused on evaluating estuaries across the biogeographic range of the species from British Columbia to Baja California (Fig 1). We started with a comprehensive list of US West Coast estuaries [Appendix A in 46]. The stakeholder team then narrowed the list by removing estuaries that are unsuitable for Olympia oysters, such as those known to have extended periods of low salinity or stagnation (i.e. bar-built estuaries, small lagoons that are closed for extensive periods). Team members from Mexico and Canada also added appropriate estuaries from their regions to the US list. For the two largest estuaries —San Francisco Bay in California, and the Salish Sea in Washington / British Columbia— regional team members delineated sub-basins that could be assessed separately (S1 Fig).

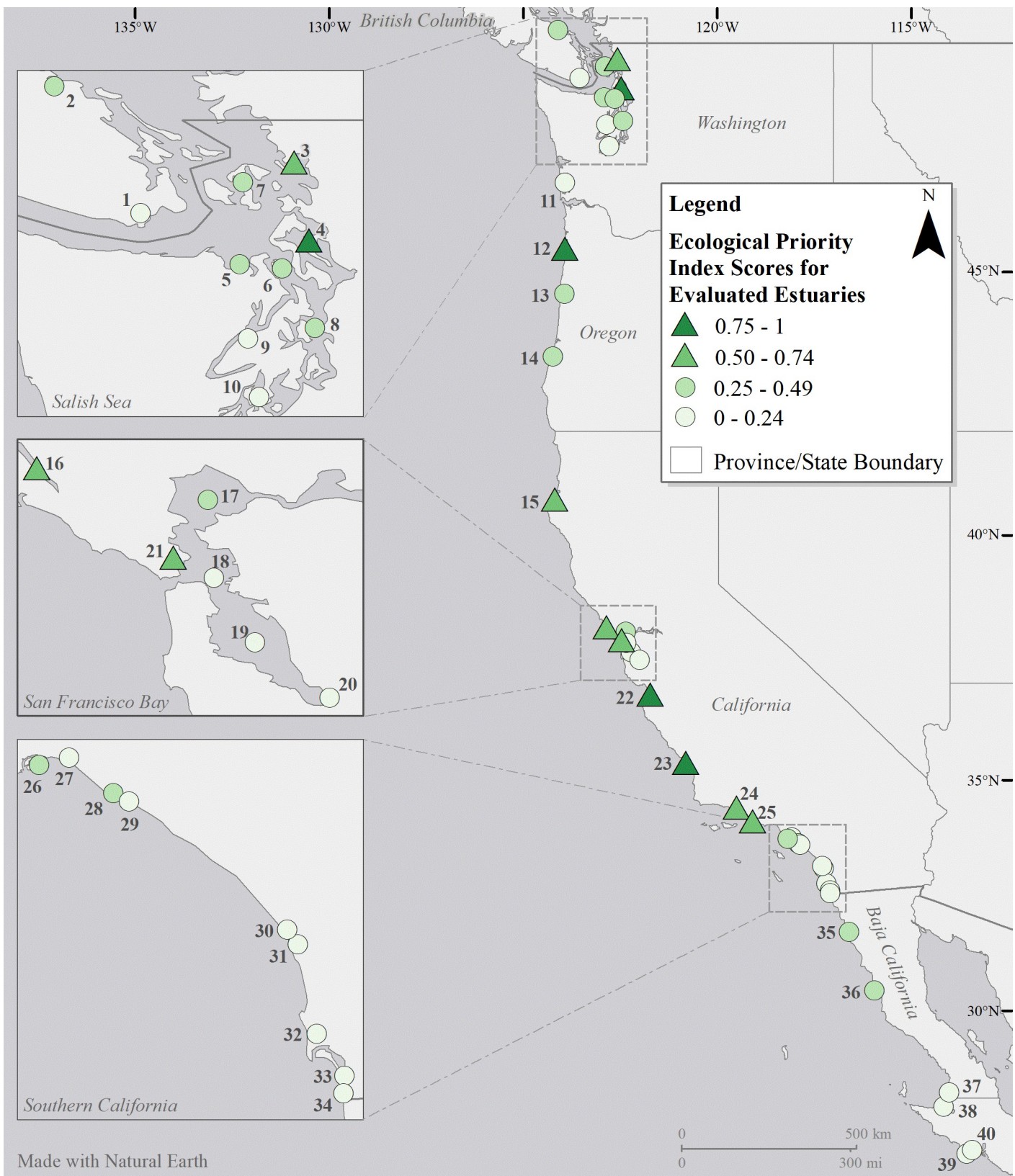

**Fig 1. Location of estuaries and their ecological priority index scores.** Names of the numbered estuaries are provided in Fig 2.

## Conservation aquaculture indices

We developed 14 criteria to assess the potential value and feasibility of new or enhanced investment in aquaculture to support conservation efforts for the native oyster across its range (Table 1). We focused our assessment on differential benefits, asking where the use of aquaculture may provide a relatively larger conservation benefit or have a greater likelihood of success. It became clear from stakeholder discussions that more information was available on spatial differences in benefits than on risks—risks such as habitat damage from gear or loss of genetic diversity could occur anywhere, and we lacked clear evidence of such risks being greater in some places than others. Both ecological and social criteria were included. These criteria were generated and refined from a longer initial list (21 were originally identified) through an iterative process with the stakeholder team; we dropped those for which there was insufficient information or that upon further consideration did not seem critical to the evaluation. The criteria were phrased as questions to clearly articulate what each criterion is addressing. We also jointly crafted a statement of rationale for inclusion of the criteria. Scoring guidelines were developed for each criterion (S2 Table).

Each estuary was assigned a score (2 = high, 1 = medium, 0 = low) for each of the 14 criteria, with higher scores indicating higher priority for aquaculture. We assigned a score of zero to fields with missing data (leading to conservative results—in the future if data are available, these scores may increase). Each estuary was scored first by the expert or experts most familiar with it, using quantitative data (e.g. on recruitment, survival, or growth from their own monitoring or publications) where available, their own expertise, and the assistance of additional local experts where appropriate. Next, multiple stakeholders worked in regional teams to review and discuss scores, provide additional relevant information and resources, and reach consensus on scores. Some estuaries are very well-studied with robust data on factors such as recruitment rates or growth rates, while others are poorly characterized. To reflect such differences, each score was annotated with a certainty level: not very certain (*), fairly certain (**), or very certain (***). To document the decision-making process in scoring, the basis for each score was briefly explained, and any relevant citations or internet resources listed; names of stakeholders that conducted the scoring for each estuary were also documented (S3 Table). It should be noted that the evaluation process described above resulted from a larger collective effort than is reflected in the author list alone (S1 Table).

To identify the places where new or enhanced investment in aquaculture could most benefit Olympia oyster populations, we developed an *ecological priority index* using only the first four ecological criteria. Our team considered the relative importance of each and decided on a weighting scheme that gave recruitment limitation the highest weight, followed by risk of extinction, and finally isolation and post-settlement mortality (Table 1). Index scores were calculated by summing the weighted scores and then dividing by the maximum possible score, so that index scores ranged between 0 (lowest) and 1 (highest). Any estuary with an index of ≥0.50 was considered an ecological priority, as this threshold is transparent and indicates that a majority of criteria were met. Estuaries that had no information for the first three criteria (recruitment, extinction risk, mortality) were initially evaluated (S3 Table) but omitted from subsequent analyses because these three criteria are fundamental to any consideration of conservation aquaculture. Our prioritization thus is conservative; in the future, if more data are available, more estuaries may emerge as priority sites for conservation aquaculture.

We then developed three additional indices: a *community restoration index*, a *community harvest index*, and a *commercial production index.* At any estuary, a conservation organization or resource management agency could potentially implement small-scale conservation aquaculture if a need to do so is identified, working with a local hatchery to outplant juveniles solely

**Table 1. Criteria used in evaluating estuaries for conservation aquaculture.** Each criterion is framed as a question, and the rationale for inclusion is provided. In the final four columns, the weighting of each criterion used to calculate the four indices is shown; if blank, this criterion was not included in the index.

| | Criterion question | Rationale for inclusion in index | Ecological Priority | Community Restoration | Community Harvest | Commercial Production |
|---|---|---|---|---|---|---|
| 1 | Is **recruitment** limited? | Aquaculture has potential to make biggest positive difference in estuaries with low recruitment. | 4 | | | |
| 2 | Is the Olympia oyster population at risk of local **extinction?** | Aquaculture has the potential to make the biggest positive difference in estuaries with very low adult populations that are at risk of disappearing. | 3 | | | |
| 3 | Is post-settlement **mortality** low? | Places with high post-recruitment survival should be prioritized, so return on aquaculture investment is maximized; avoid sites with high mortality due to high drill predation, freshwater events, etc. | 2 | 2 | 2 | 1 |
| 4 | How **isolated** is this Olympia oyster population? | Estuaries that are geographically and/or genetically isolated and do not have larval exchange from nearby populations are more vulnerable to a potential loss of an entire population, and may benefit more from aquaculture as a tool. | 2 | | | |
| 5 | Is there a nearby **hatchery** that has produced Olympia oyster spat from local broodstock using conservation protocols? | A hatchery that is in the same estuary will reduce risks associated with shipping spat from greater distances including introducing pathogens, parasites, or non-native fouling species. Conservation protocols are aimed at retaining local genetic adaptations and diversity. | | 1 | 1 | 2 |
| 6 | Is it **safe to eat** shellfish? | Only areas where water quality allows for safe consumption of shellfish can be used for commercial production or community harvesting. | | | 3 | 3 |
| 7 | Do **regulations allow harvest** of Olympia oysters? | Areas where harvest is allowed should be prioritized for community groups interested in harvest. | | | 2 | |
| 8 | Is post-settlement **growth** of Olympia oysters high? | High growth rates reduce the length of time between outplant and harvest, relevant to community harvest groups and commercial growers. | | | 1 | 1 |
| 9 | Are Olympia oysters or any other species of **bivalves** currently being **farmed** in this estuary? | New or expanded Olympia oyster aquaculture by community groups or commercial growers is facilitated if there is existing infrastructure for farming bivalves, and a track record of safe water quality, harvest, etc. | | | 1 | 1 |
| 10 | Is local Olympia oyster restoration/ enhancement part of the **management or conservation plan** of any organizations? | Aquaculture investment and permitting is facilitated where native oyster restoration is identified as a priority by multiple organizations. | | 2 | | |
| 11 | Are there **community** or volunteer groups currently engaged in **Olympia oyster** restoration? | Aquaculture-based restoration is more likely to succeed and positively affect people where there are engaged communities. | | 2 | | |
| 12 | Are other **community** groups currently engaged in restoration of other species of **bivalve/shellfish**? | Existing groups engaged in restoration with similar species could indicate a capacity for or interest in incorporating Olympia oyster restoration in the future. | | 1 | | |
| 13 | Are there **Native American Tribes or First Nations** currently engaged in **bivalve/shellfish** restoration or wild harvest? | Indigenous stewardship increases the chances of successful long-term restoration and management of oyster populations, and increases the priority of doing so, to sustain a legacy of cultural practices. | | 1 | 1 | |
| 14 | Are other **community** groups (non-commercial) growing shellfish (oysters or other species) for **harvest?** | Existing groups that grow and harvest shellfish can increase the success of restoration efforts via the stewardship, maintenance and management of oyster beds. | | | 1 | 1 |

to enhance natural populations. However, we were interested in also addressing the suitability of each estuary for three other types of conservation aquaculture projects: 1) community restoration (restoration involving hatchery-raised oysters that is primarily driven by local community interests or engagement), 2) community harvest (deployment of hatchery-raised oysters specifically in order to be harvested and consumed by the local community), and 3) commercial production (aquaculture of oysters raised by growers and sold for profit). The indices for all three of these project types included one ecological criterion, post-settlement mortality, because engaging in novel aquaculture endeavors is likely to fail in places with high mortality. Each of the three indices used a different suite of other criteria, weighted differentially (Table 1), as determined by discussion among the stakeholder team. The indices were calculated summing weighted scores and then divided by the maximum possible value (to obtain scores from 0–1) with two exceptions. For community harvest, an estuary automatically received an index score of zero if it had received a zero for safety of shellfish consumption or legality of Olympia oyster harvest, since either of these preclude community harvest. For commercial production, an estuary automatically received an index score of zero if it had received a zero for safety of shellfish consumption. Again, estuaries were deemed a high priority for consideration of a particular project type if the index score was ≥0.50.

## Results

### Rewards and risks of conservation aquaculture with Olympia oysters

Initially, the stakeholder team identified 32 different potential rewards and 29 potential risks associated with conservation aquaculture of Olympia oysters. These were aggregated and refined to 12 rewards and 11 risks that had direct relevance solely to conservation aquaculture projects (e.g. not general rewards that would also be associated with oyster restoration, such as water filtration). Social, cultural, and economic benefits from conservation aquaculture were the most represented benefits (67%), compared to the ecological returns (33%). The risks were generally more varied: ecological risks were represented most strongly (55%), followed by economic (27%), and then social and cultural (18%). Each benefit or risk was assigned to one of the categories we identified based on the primary beneficiary: Olympia oyster populations, the coastal ecosystem, conservation practitioners, Tribes / First Nations, local community members, and commercial growers.

For Olympia oyster populations, the primary benefit of utilizing conservation aquaculture was the potential to dramatically increase local population numbers fairly rapidly, particularly in recruitment-limited estuaries (Table 2). The primary risks to populations involved genetic concerns associated with using hatchery-reared oysters, including reduced genetic diversity and loss of local adaptations, and the potential for the density-dependent emergence and spread of diseases within native oyster populations (Table 3).

While stakeholders originally identified many benefits for the coastal ecosystem derived from Olympia oyster populations, only two were specific to projects employing aquaculture, providing additive value beyond traditional (non-aquaculture) Olympia oyster restoration. These potential benefits were the reduced introduction of non-native fouling species associated with the culture of native oysters versus introduced oysters, and the benefits of additional (artificial) habitat provided by aquaculture gear, particularly in areas such as unstructured mudflats (Table 2). Potential risks included negative ecological consequences associated with the transfer of oysters from within hatcheries into wild populations and potential negative alterations of the coastal habitat as a result of the use of commercial farming via aquaculture gear (Table 3).

For conservation practitioners, conservation aquaculture techniques may support the management of oyster populations for future climate conditions, since they enable the search for

**Table 2. Potential rewards of conservation aquaculture of Olympia oysters, organized by primary recipient group.** Only those rewards uniquely associated with conservation aquaculture, not with oyster restoration through any means (e.g. water filtration), are presented.

| Reward to (primary recipient) | Benefit of Olympia Oyster Conservation Aquaculture |
|---|---|
| Olympia Oysters | Scaled or rapidly increased population numbers where there is low recruitment or recruitment failure for populations that are severely declined, at risk of local extinction, and/or isolated from other populations |
| | Increased genetic diversity in small populations, where genetic bottleneck/ allee effects are likely to occur |
| Coastal Ecosystems | Increased structure and habitat for other fish and marine invertebrates from aquaculture gear |
| | Reduced introduction of non-native fouling species if fewer non-native oysters are commercially grown, if native oysters become commercially viable |
| Conservation Practitioners | Ability to manipulate reproduction, test tolerances to environmental conditions (e.g. to search for phenotypes more resilient to climate change effects) |
| | Leveraging private industry for conservation gain through partnerships with commercial growers |
| Tribes / First Nations or Local Community Members | Increased community engagement with, knowledge of, and/or stewardship of coastal ecosystem through consumption/harvest of a local food source made possible at a larger scale through aquaculture |
| | Additional revenue streams (e.g. ancillary downstream businesses) for waterfront and community |
| | Increased traditional food source benefit |
| | Maintained social/cultural continuity of traditional food gathering |
| Commercial Growers | Potential to improve perceptions of aquaculture and/or increase interest in native aquaculture species |
| | Diversification of, increased income, and increased resilience (climate or other) for grower portfolio |

**Table 3. Potential risks of conservation aquaculture of Olympia oysters, organized by primary recipient group.**

| Risk to (primary recipient) | Risk of Olympia Oyster Conservation Aquaculture |
|---|---|
| Olympia Oysters | Reduced genetic diversity of hatchery-raised animals vs. wild population |
| | Reduced local adaptation and/or plasticity |
| | Increased disease emergence and/or spread; increase of parasites with increasing population densities |
| | Increased risk of poaching if hatchery production raises profile of native oysters and increases awareness of wild or restored oyster locations |
| Coastal Ecosystems | Increased disease or pests spread from Olympia oysters to nearby native species or habitat |
| | Negative alterations of the natural habitat from aquaculture gear, including increased plastics in the marine environment |
| Conservation Practitioners | Difficulty in creating BMPs at appropriate scales and/or risk of growers producing or selling Olympia oysters without adhering to BMPs and protocols |
| | Greater competition between industry and conservation groups for funding and/or resources |
| Tribes / First Nations or Local Community Members | Disempowerment by not taking into account local community priorities or restricted tribal areas |
| Commercial Growers | Riskier and less profitable species to raise than non-native species due to slow growth/longer time to harvest |
| | Increased abundance of Olympia oysters from aquaculture could lead to increased larval production and settlement on other cultivated species, with negative results for both other species and Olympias in being removed from water |

phenotypes that are more resilient to environmental stressors. Conservation practitioners may also benefit from engaging new partners and leveraging the private industry (that are using their own resources for commercial oyster production) for conservation gain (Table 2). Two risks involved the creation and adherence to conservation-centered better management practices (BMPs): the difficulty in creating (BMPs) at appropriate scales for commercial hatcheries, and the potential for some in the commercial oyster industry to produce Olympia oysters without adhering to protocols that further the goals associated with conservation aquaculture. An additional risk was the potential for greater competition between industry and conservation groups for funding and/or resources (Table 3).

For Tribes, First Nations, and community groups, a primary benefit of conservation aquaculture projects was the harvest and consumption of a native, locally sourced food in estuaries where wild Olympia oyster populations cannot sustain a harvest. For communities of Indigenous Tribes and First Nations in particular, benefits also included increased access to a traditional food source and the maintenance of social/cultural traditional food practices (Table 2). However, we also identified the potential for lack of early consultation with Tribes and/or First Nations as a primary risk which could lead to disempowerment if the management and stewardship priorities of these communities are not taken into account prior to the start of a project (Table 3).

Finally, for commercial oyster growers, adding Olympia oysters to complement existing farmed species can provide diversification of the grower's portfolio, which may be key to continued commercial viability. Additionally, conservation aquaculture of native Olympia oysters can provide benefits related to improving public perceptions of aquaculture and potentially result in increased interest in cultured seafood (Table 2). Risks included Olympia oysters being a generally less profitable and often riskier species to raise due to their slow growth, and the potential of Olympias bio-fouling or settling on other cultivated species (Table 3).

## Priority estuaries and project types

We identified 66 estuaries as suitable for Olympia oyster populations throughout the species' range along the west coast of North America (with sub-basins of the Salish Sea and San Francisco Bay treated as estuaries). Each of these estuaries was scored for all 14 conservation aquaculture criteria, with scoring rationale and references clearly documented (S3 Table). Of these, 40 estuaries had sufficient data for the first three ecological criteria, which were vital for evaluating the value of conservation aquaculture (Figs 1 and 2). The remaining 26 estuaries had insufficient information and were not considered further. Overall, more estuaries at both the northern and southern ends of the distribution were omitted due to limited available data, and the remaining estuaries that were retained in British Columbia and Baja California still had many "unknown" scores for individual criteria. Thus, our results are likely conservative given the data limitations (scores of 0 for missing data will later be increased as data become available).

Of the 40 estuaries with sufficient data, ten estuaries were identified as ecological priorities (Figs 1 and 2) for new or increased investment in conservation aquaculture of Olympia oysters. Two of these estuaries (Netarts Bay in Oregon and Elkhorn Slough in California) received the highest possible index score (1.0). Three other estuaries scored quite high (>0.7): Whidbey Basin in Washington and Morro Bay and Carpinteria Marsh in California.

The ten ecological priority estuaries varied in the appropriateness of different project types (Fig 2, S2 Fig). Half of these estuaries ranked highly for community restoration (five estuaries = >.50), and 40% ranked highly for community harvest and commercial production each (four estuaries = >.50 for each project type). Four of the ecological priority estuaries—all

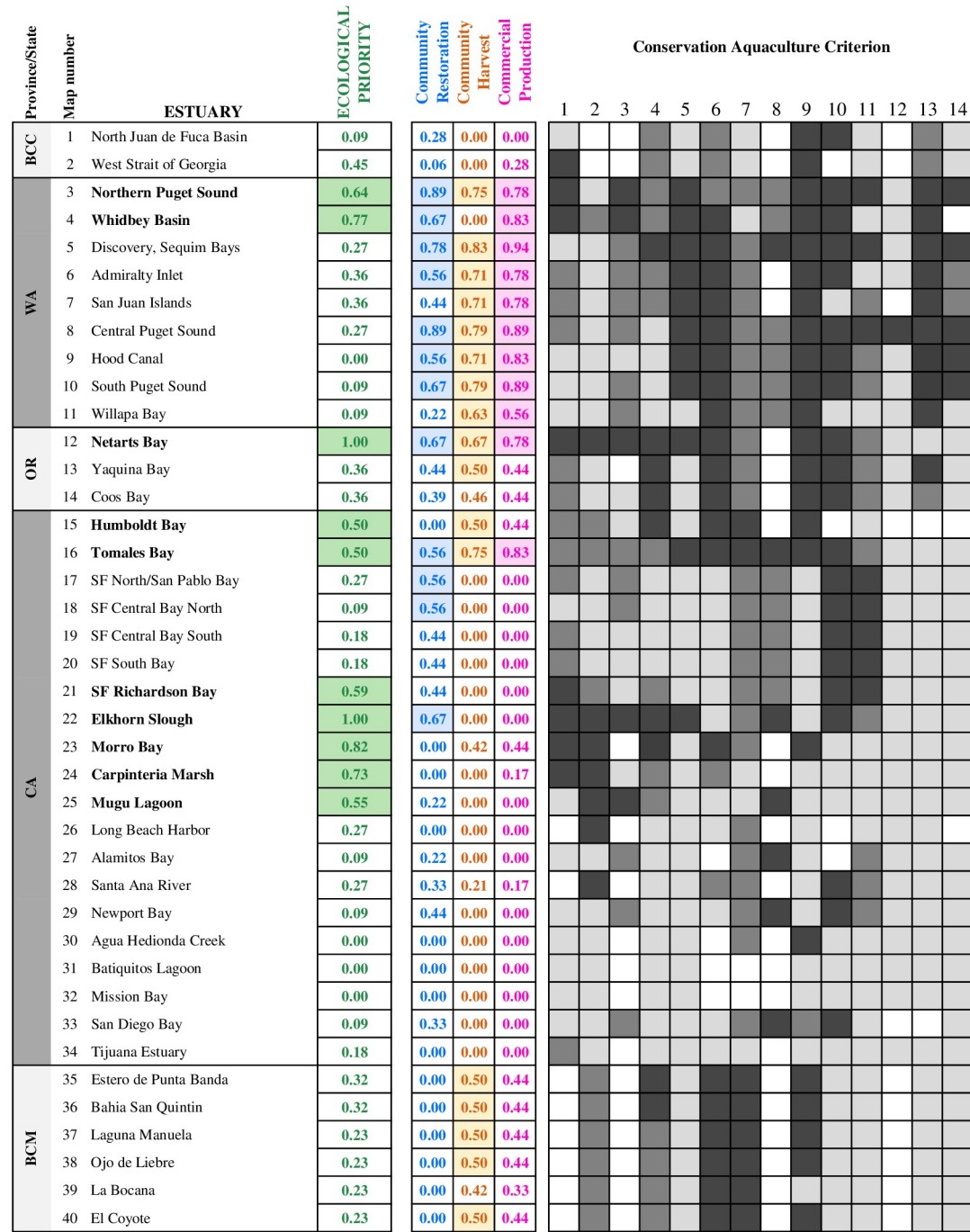

**Fig 2. Conservation aquaculture indices.** Estuaries are arranged from North to South, with the exception of subbasin areas, which are grouped for simplicity. Province or state abbreviations are shown (BCC = British Columbia, Canada; WA = Washington, OR = Oregon, CA = California, USA; BCM = Baja California, Mexico). The names of the ten estuaries that emerged as ecological priorities are shown in bold font; all index scores ≥0.5 are highlighted. The individual scores for each criterion are shown to the right (darker shading represents higher scores; missing data shown in white).

within the central range of the species from San Francisco Bay to Mugu Lagoon—did not rank highly for any of the three project types, and are instead currently best suited for projects led by conservation or resource management organizations.

## Discussion

Our assessment of conservation aquaculture for Olympia oysters is timely and highlights the need to consider this strategic approach to mitigate the steep declines faced by other marine species globally. We have developed and implemented a transparent and analytical framework (Fig 3) designed to evaluate conservation aquaculture as a tool to support imperiled species, by linking specific rewards and risks with particular end-user groups. Throughout the planning process, we identified ways in which conservation aquaculture can address both current management challenges (e.g. recruitment failure), and advance forward-thinking strategies for future challenges (e.g. resilience of oysters to the effects of future climate change scenarios). This type of assessment is needed for other foundation species, especially where aquaculture is already being used as a tool to address declines (e.g. reef-building corals).

### Tradeoffs and risk mitigation of conservation aquaculture

Our collaborative process engaging diverse stakeholders identified the key rewards and risks of conservation aquaculture for Olympia oysters. Below, we discuss these by end user or beneficiary, and make recommendations to minimize these risks.

**Olympia oyster populations.** Any successful restoration directly benefits the focal species that is being restored. As such, restoration efforts aimed at rebuilding Olympia oyster populations can be beneficial for restoring numbers and distribution towards historical baselines, and in very threatened populations, for preventing local extinction. What makes restoration via conservation aquaculture different from other restoration approaches is the potential to dramatically increase local population numbers fairly rapidly. In recruitment-limited estuaries, this cannot be achieved simply by providing bare settlement substrates, which is the most common restoration approach for Olympia oysters [28]. Supplying thousands of hatchery-raised juveniles can boost recovering populations to a threshold where they can become self-sustaining, as has been accomplished in Sequim Bay and Fidalgo Bay in Washington [47].

The primary risks identified by our stakeholder team involved genetic concerns associated with using hatchery-reared oysters, including reduced genetic diversity and loss of local adaptations [48]. Olympia oysters have locally differentiated population structure [49, 50] and have been shown to exhibit local adaptations for the timing of reproduction [51] and tolerance to salinity fluctuations [52]. Maintaining such local adaptations when possible is important, thus we recommend the use of responsible conservation aquaculture protocols such as those developed for the Chew Center by Puget Sound Restoration Fund and Washington Department of Fish & Wildlife (e.g. [31]). These include the use of large numbers of local adult oysters for broodstock, and techniques specifically aimed at reducing hatchery selection. Considering hatchery conditions is also critical since Olympia oysters have been shown to have strong carryover effects on reproduction relative to environmental conditions [53].

For extremely small populations, especially those facing local extinction, the benefit of increasing the population size may outweigh genetic concerns such as outbreeding depression [41]. Introducing genetic diversity into these populations may also be beneficial now, and in the future as environmental conditions change due to climate change [54]. Our ecological priority index resulted in recommendations for increased use of this tool primarily in estuaries with very small populations, where potential genetic risks are likely overshadowed by genetic benefits and decreased extinction risk.

The risk of density-dependent emergence and spread of diseases within native oyster populations was also identified as a concern. Currently, diseases and pathogens do not appear to play a major role in Olympia oyster mortality [55]. However, higher densities in the hatchery

## EXPLORE NEEDS AND FEASIBILITY

| | |
|---|---|
| **1) Determine whether the <u>species requires restoration</u>** | Have populations declined? Are numbers lower than historically? |
| **2) Assess whether <u>reproduction is limiting</u> population growth** | Is recruitment a main factor in low population growth at least in some places? |
| **3) Evaluate whether <u>aquaculture is feasible</u>** | Has this or a related species been raised in hatchery? Could it be? |
| **4) Assemble <u>experts and stakeholders</u> with which to complete all subsequent steps** | Including aquaculture experts, conservation scientists, and communities benefiting from services |

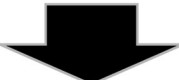

## IDENTIFY AND WEIGH TRADE-OFFS

| | |
|---|---|
| **5) Brainstorm with stakeholders and experts on <u>potential risks and rewards</u>** | Draw on data, expert opinion, personal experience with target or similar species as available |
| **6) <u>Prioritize</u> most important trade-offs** | Group and evaluate them |
| **7) Determine whether <u>rewards outweigh risks</u> for this species** | (At least in some places) |

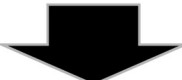

## PRIORITIZE LOCATIONS AND PROJECT TYPES

| | |
|---|---|
| **8) Identify <u>possible locations</u> for projects** | Where species has declined, or conditions are suitable for future populations |
| **9) Develop <u>criteria for evaluating net benefit</u> to the species of aquaculture in each location** | Incorporate risks and rewards identified above |
| **10) Develop <u>criteria for assessing suitability of different project types</u>** | Production by conservation organization vs. commercial growers, etc. |
| **11) Design <u>indices</u> that combine information from multiple criteria** | Summary calculation may weight some criteria more than others |
| **12) <u>Score all locations</u> for the criteria and indices** | Priority locations and suitable project types emerge |

**Fig 3. Conceptual diagram of steps to take in evaluating conservation aquaculture for a new species or region.** The diagram follows a logical chronological flow, but in practice some steps may occur simultaneously or there may be iterative rounds revisiting particular steps. For the first steps, the process should only move forward if the determination of the previous step is affirmative (e.g. only move to step 2 if the species requires restoration, to step 3 if reproduction is deemed limiting, to step 4 if aquaculture is feasible, to step 5 if there is a team, etc.).

or estuary can increase this risk [56]. Careful monitoring for pathogens, and avoidance of transfers among hatcheries, should be used to manage this risk.

**Coastal ecosystems.** Olympia oyster restoration through any means provides benefits to the ecosystem, such as increasing the diversity of benthic-water column coupling and filter feeders [57] and enhancing habitat for other fish and marine invertebrates [21]. While our stakeholder team did not identify many ecosystem-wide benefits exclusive to aquaculture-based restoration, these general benefits also result from conservation aquaculture projects. One potential reward specific to conservation aquaculture is the reduced introduction of non-native fouling species, which include parasites and pathogens associated with culture of non-native species in distant hatcheries, versus native species in local ones. However, this benefit would only accrue if Olympia oysters become commercially viable and production is wide-spread, resulting in concurrent decreases in aquaculture of non-native species. In addition, the artificial structure provided by aquaculture gear in areas such as unstructured mudflats can provide similar benefits for fish and invertebrates as other 'artificial reefs' [8, 58].

However, any aquaculture endeavor involves some level of risk of transferring non-native species, including pathogens and parasites. There are potential negative ecological consequences of conservation aquaculture associated with the transfer of oysters from within hatcheries into wild populations, although disease risks appear modest [55]. In addition, negative alterations of the coastal habitat can occur as a result of the use of commercial farming via aquaculture gear, for example by possibly contributing to microplastics in the estuaries where oysters are grown commercially [59]. Aquaculture gear also has the potential to damage particularly sensitive estuarine habitats or create space conflicts with other declining marine foundation species in need of protection, such as seagrasses [5, 60]. We recommend that native oyster culture take place within the existing footprint of commercial operations when possible to minimize negative impacts on natural estuarine habitat that can occur on small spatial scales as the result of shading and increased sedimentation. It is worth noting that at larger scales, there may be positive effects of aquaculture gear, due to increased water column nutrients for seagrasses and structural refugia for fish and invertebrates provided by gear in unstructured mudflat habitats [61–63]. We recommend that the overall positive and negative impacts—and the spatial scales of those impacts—be carefully assessed in the deployment of commercial aquaculture gear.

**Conservation practitioners.** Conservation aquaculture represents a relatively new and unique tool for practitioners to use for the enhancement or rebuilding of Olympia oyster populations where traditional restoration practices have not been successful or well-funded. As climate change increasingly affects coastal ecosystems, conservation practitioners may also use aquaculture techniques to search for phenotypes more resilient to climate change effects (e.g. those with higher tolerances to changed or predicted climate-driven environmental conditions), as has been done with other species like corals [64]. Despite its promise, due to the high degree of uncertainty about climate effects on Olympia oysters [65], we recommend further study before we can explicitly support assisted evolution approaches with this species.

Conservation practitioners can also benefit by engaging new partners and leveraging private industry funds and resources from commercial oyster production for conservation purposes. Considerable benefits may be realized despite the limited public funding for conservation in general and for restoration of the Olympia oyster in particular. However, there

are risks to conservation practitioners engaging in these new partnerships. One is the difficulty of creating conservation-centered best management practices (BMPs) at appropriate scales that can or will be followed in commercial hatcheries. A related risk is the potential for some in the commercial oyster industry to produce and sell Olympia oysters without adhering to protocols that further the goals associated with conservation aquaculture. Finally, there is potential for greater competition between industry and conservation groups for funding and/ or resources.

To address the risks associated with industry partnerships, we recommend that conservation practitioners work with hatcheries that have clear, appropriate conservation practices and protocols in place, and that they work with commercial growers to understand the particular challenges and opportunities faced by hatcheries following these conservation BMPs, to ensure that all parties continue toward shared goals. We also recommend conservation practitioners have transparent conservation-first policies in place regarding their rationales and conditions in which they partner with industries (e.g. [66]).

**Tribes, first nations, and community groups.** Where populations of oysters cannot sustain a wild harvest, conservation aquaculture projects can provide some specific benefits through the harvest and consumption of Olympia oysters, including connecting people with coastal ecosystems through the enjoyment of a native, locally sourced food. This is evidenced by the recent increase in interest in Olympia oysters in the Pacific Northwest and California as part of the popular "slow food" movement [67]. Rebuilding populations of Olympia oysters can also encourage an increased engagement with, knowledge about, and/or stewardship of coastal ecosystems within communities that may otherwise lack connection with native oysters as a component of West Coast estuaries. For communities in Indigenous Tribes and First Nations in particular, conservation aquaculture projects can increase access to a traditional food source and serve to maintain social/cultural continuity of traditional food practices [68, 69].

However, our group identified the potential for lack of early consultation with Tribes and/ or First Nations as a primary risk which could lead to the disempowerment of these communities if the management and stewardship priorities for tribally ceded areas are not taken into account prior to the start of a project. Further risk ensues when restricted tribal properties are accessed or prioritized for conservation aquaculture by those outside of the community without the appropriate prior consultation and consent from the Tribe or First Nation. While Indigenous Tribes, First Nations, and local communities were combined into one user group in this analysis due to the overlap of rewards and risks identified, it should be noted that Indigenous Tribes and First Nations are not stakeholder groups, but sovereign Governments that do not always share the same values, nor are governed by the same regulations, as their non-Indigenous neighboring communities.

We recommend the continuation and expansion of engaging communities in conservation aquaculture projects for restoration and harvest wherever possible. It is critical that any organization partnering with Indigenous Tribes and First Nations involve those communities directly from the start of the planning of a project and respect tribal access to, applicable legal authorities over, and stewardship of traditional lands. This includes asking permissions to access and conduct projects on land where appropriate and engaging in a collaborative process to ensure that the community's priorities for resource management, cultural heritage, and stewardship of the land are helping drive the project goals. We also recommend that conservation aquaculture projects that involve harvest include a component of public education about not harvesting or disturbing wild Olympia oyster populations that are on restoration sites or in non-harvestable areas.

**Commercial growers.** For commercial oyster growers, adding a species such as the Olympia oyster to complement existing farmed species can provide diversification to the grower's

portfolio. With the increasing impacts of climate change, and an often unpredictable marine environment [70, 71], diversification may be key to continued commercial viability. As with other fisheries, having a more diverse portfolio of oyster species could create additional economic opportunities or buffer against economic loss [72]. For example, a recent outbreak of oyster herpesvirus (OsHV-1) in the *Crassostrea gigas* population in Southern California required the destruction of a season's worth of infected oysters, which could not be sold, representing significant financial losses for growers [73]. Olympia oysters are not as vulnerable to herpes, or many of the other diseases that plague other commercial species, including *Crassostrea* [55], and show relatively increased resilience to climate effects [65], thus making them ideal to add to the grower's portfolio to buffer against such losses. Olympia oysters also provide an entirely different flavor profile than Pacific oysters, and represent a locally sourced native species, which could provide additional marketing and market differentiation opportunities [42].

Engaging in conservation aquaculture of Olympia oysters can also provide potential benefits related to public perceptions of aquaculture. In addition to farmed bivalves being one of the lowest-impact forms of marine aquaculture and ways of producing animal protein [74], there is emerging research on the positive ecosystem services that "restorative" commercial shellfish and seaweed aquaculture can provide for water quality, habitat structure, and climate resilience [8, 61, 75]. Despite this, there remain some negative associations in the US with farmed seafood and marine aquaculture. By focusing on growing a native species and for a stated benefit of increasing native species populations, commercial growers can engage the public under an explicit conservation framework and potentially see increased interest from or improved public perceptions of farmed seafood.

While diversification and perceptions are important, it must be noted that Olympia oysters are generally less profitable, due to their smaller size and longer growing period, and therefore a riskier species for farmers to grow. Also, if Olympia oysters are grown in close proximity to other species, there is the risk that larval overflow could increase "bio-fouling" when Olympia oyster larvae settle on other cultivated species [76], with negative results for both the other cultured species and the Olympias oysters that are removed from the estuaries during harvest.

We recommend engagement of conservation aquaculture by commercial oyster growers in priority areas where: the ecological benefits outweigh the risks; the grower has enough internal financial capacity to support the growing of a species that can take 1–2 years longer to mature and may be initially less profitable than other oyster species; and there is a desire to grow Olympia oysters for both commercial and non-commercial reasons. While it is currently not as profitable as other species, the commercial growers in our expert group that choose to focus on growing and providing Olympia oysters to the public were doing so not only for the market differentiation and potential future profits, but with historical and cultural education benefits, and cultural continuity in mind.

## Geographic prioritization with conservation aquaculture indices

We identified ten estuaries along the entire biogeographic range of the Olympia oyster where new or increased investment in conservation aquaculture is a high priority. Application of the ecological priority index provided a transparent, robust approach to selection of these priority estuaries from 66 estuaries that were evaluated in a consistent manner. The Native Olympia Oyster Collaborative (https://olympiaoysternet.ucdavis.edu/) was formed to provide such regional perspectives and syntheses, complementing the on-the-ground restoration work which is largely place-based and locally driven, by individual members of the Collaborative. For funders working at a broad spatial scale (e.g. The Nature Conservancy, Pew Charitable Trusts or NOAA), identification of the ten priority estuaries will help direct future grants for

conservation aquaculture to the places where the return on investment is highest. For regulatory agencies, a robust process conducted regionally but drawing on local knowledge will help to inform the permitting process for the highest priority estuaries. While our evaluation was conducted across the full range of the species, the tool we developed can be applied at any scale, for instance for selecting sites within an estuary. We have provided a spreadsheet version (S3 Table) with weighting and formulae so other end users can customize it for their needs. Our tabular approach complements GIS-based decision-support tools developed to prioritize oyster restoration sites [75, 77].

**Ecological priority areas.** The ecological priority index was designed to prioritize sites where the rewards of aquaculture are likely to greatly outweigh the potential risks. The ten estuaries that emerged as ecological priorities thus represent locations where our stakeholder team can confidently recommend consideration of increased investment in aquaculture as a tool to support native oyster conservation. We weighted recruitment limitation most highly among ecological criteria, since oyster populations where estuary-wide recruitment failure is common (no successful reproduction anywhere in the estuary in many years) stand to benefit the most from enhancement of reproduction through aquaculture. Risk of local extinction due to small or declining population size, and high isolation resulting in lack of larval transport from other populations were also critical determinants of ecological priority. The other key requirement was high post-recruitment survival, since investment in aquaculture is not merited if all outplanted juveniles die.

The two estuaries that scored highest as ecological priorities for conservation aquaculture (Netarts Bay in Oregon and Elkhorn Slough in California) are places where estuary-wide recruitment failure is common, because current populations are tiny due to past declines, and because larval retention is challenged by low residence time resulting from strong tidal currents [78]. Both are in areas where genetic analyses suggest isolation is high, with limited connectivity to other populations [50]. Restoration incorporating aquaculture has been attempted in both places [36, 79]. We recommend scaling up these efforts until populations there are self-sustaining, large enough, and located in areas of higher residence time so that successful recruitment occurs at least in some locations in the estuary in most years.

While we identified ten sites along the coast that represent high priorities for future investment, it is clear that further consideration of the feasibility and potential benefits of conservation aquaculture is required at a local or regional scale before proceeding with new projects. Our index used four critical criteria to identify these priority sites, but at a smaller spatial scale, much more information can be used to inform decision-making. Such additional information might lead to local adjustment of prioritization. For instance, Hood Canal in Washington did not emerge as a priority in our index, but conservation aquaculture projects are being considered in the lower portion of the Canal because this location appears to be recruitment limited, other restoration methods may not prove effective in this area, and because there is strong local support for boosting populations. Conversely, Richardson Bay in California did emerge as a priority, but local practitioners have indicated that invasive oyster drills pose a strong threat in many locations, and risks from the predatory drills must be mitigated before proceeding with aquaculture-based restoration at sites with high densities of drills.

**Project types.** The project type indices reveal which approaches are currently possible at each site. At every estuary, a conservation organization or resource management organization could lead a conservation aquaculture project for Olympia oysters. Such projects typically engage community members as volunteers or in outreach, but are not community-driven—the design and coordination of the project is led by paid staff from the conservation organization. An example of this is a recent aquaculture-based restoration project at Elkhorn Slough in central California, led by the Elkhorn Slough National Estuarine Research Reserve (Fig 4A).

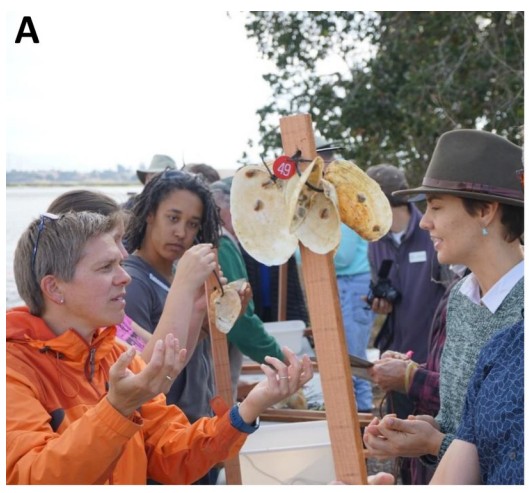
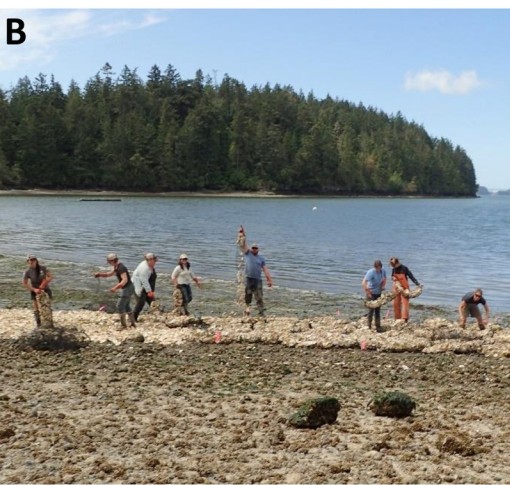
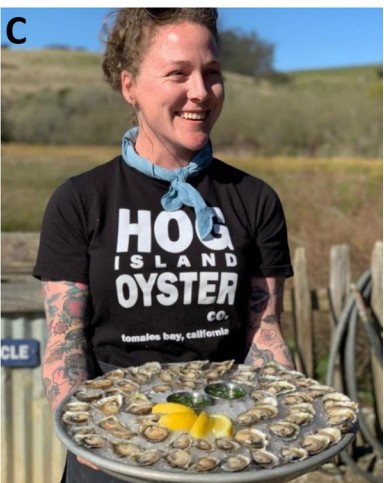

**Fig 4. Examples of different approaches to implementing conservation aquaculture with Olympia oysters.** A) Restoration with hatchery-raised juveniles led by a coastal management organization. Staff of the Elkhorn Slough National Estuarine Research Reserve, with community volunteers and partners, assemble stakes with clam shells bearing hatchery-raised juveniles. (Photo: B. Tougher). B) Community restoration. Staff members from the Swinomish Indian Tribal Community work with AmeriCorps volunteers from the Nooksack Salmon Enhancement Association to enhance habitat for Olympia oysters on Swinomish tidelands. (Photo: J. Barber); C) Commercial production and sale of Olympia oysters (Photo: M. Wilkinson, Hog Island Oyster Company). All individuals shown in images provided prior consent.

The capacity to implement other types of projects is site dependent, and currently more limited across estuaries. Restoration led by the community rather than by conservation organizations or resource management agencies appears a viable approach in many places, and currently ranks high as a potential project type at half of the ten priority estuaries (Fig 2). Community restoration depends on leadership by local individuals or groups with strong interests in growing oysters for restoration purposes. For example, in South Puget Sound, tideland owners are passionate about supporting Olympia oysters, and purchase spat to grow juveniles on their tidelands as "shellfish gardens". In North Puget Sound, the Swinomish Indian Tribal Community is engaged in restoration of oyster habitat as a cultural legacy and to support other species such as salmon (Fig 4B).

Community harvest involves local individuals or groups growing oysters specifically for harvest and human consumption, and ranks high as a potential approach at four of the ten priority estuaries (Fig 2). When community harvest occurs in ecological priority areas and contributes to supporting non-harvested components of the population in the estuary through larval spillover, we consider the practice a part of conservation aquaculture. This type of project has promise for Olympia oysters, but is mostly still in the concept stage; we do not know of any examples of successful projects of this type to date. The slower growth rate and lower availability of hatchery-raised spat makes community harvest projects with the native oyster species more challenging than projects involving the non-native Pacific oyster. Various Tribes in the Puget Sound area and Oregon are interested in restoration of the native oyster for eventual harvest, but since populations are so depleted near Tribal communities, the first step is to rebuild healthy oyster beds, with harvest still an aspirational future goal.

Commercial aquaculture projects also show potential for supporting oyster populations, and ranked high in four of the ten priority estuaries (Fig 2). If conducted in ecological priority areas using local broodstock, a nearby hatchery, and conservation protocols, this can contribute to conservation and restoration by generating larvae which support the wild population in the area. Various small-scale growers produce Olympia oysters as a part of their portfolio and implement appropriate protocols (Fig 4C). Humboldt Bay and Morro Bay both have commercial aquaculture operations for non-native Pacific oysters; if there were a hatchery nearby

producing Olympia oysters from local broodstock, these would also rank as high priorities for Olympia oyster commercial production.

**Regional recommendations.** Our analysis revealed strong regional differences in knowledge and capacity for implementing conservation aquaculture projects, which could be decreased through strategic funding and planning efforts. The information available on Olympia oysters varies substantially across the species' range (i.e. variations in missing data in Fig 2). Information needed to assess the benefits or risks of restoration is particularly lacking at both the southern and northern ends of their distribution. In Baja California, additional funding could help to fill critical data gaps and allow more thorough exploration of restoration needs and potential. In British Columbia, oyster restoration is not currently a governmental management strategy; current management goals are met through implementing the Species at Risk Management Plan for the Olympia Oyster, including surveys to assess shifts in abundance and restrictions on harvest.

The engagement of Indigenous Tribes in oyster restoration and harvest only in Washington and Oregon (high scores for criterion 13 only in top/northern section of Fig 2) is another striking regional pattern. Indigenous Tribes and First Nations harvested oysters for millenia along the entire range of the Olympia oyster [22, 23, 80]. Where possible given regional policy and regulations, conservation organizations and resource management agencies could work to engage Indigenous communities that are interested in collaborative efforts to restore native oyster harvests as a traditional food source and cultural practice. However, this could prove challenging where Indigenous people no longer have access to estuaries, or where water quality is so poor as to preclude consumption of shellfish, as is the case in heavily urban and agricultural areas of central and southern California.

Finally, the distribution of hatcheries that could produce native oysters and facilitate aquaculture-assisted efforts is also centered in the Pacific Northwest. Indeed, to our knowledge no hatchery has produced Olympia oysters south of Elkhorn Slough in central California—leaving about 1500 km of the southern range of the species without current hatchery capacity. We recommend that such capacity be advanced in coming years, perhaps by university marine laboratories, particularly somewhere near the three southern California ecological priority sites (Morro Bay, Carpinteria, Mugu Lagoon).

## Conclusions

The new approach and assessment tools developed to evaluate conservation aquaculture for Olympia oysters provide a template that is clearly needed to evaluate the potential use of conservation aquaculture for other declined species, and especially in evaluating the tradeoffs between rewards and risks where aquaculture is already being used as a tool to address declines (e.g. reef-building corals). Initiatives are being undertaken globally at many international conservation organizations to engage in conservation aquaculture, and our evaluation can contribute to shaping the use of conservation aquaculture as a tool worldwide. We take an ecological approach that centers on the measurable benefits that conservation aquaculture can uniquely provide to at-risk populations. We incorporate broad stakeholder involvement from the start of the process to develop an explicit risk/reward framework that facilitates decision-making for different goals such as commercial production, community restoration, and community harvest. Our work illustrates how to create and leverage existing partnerships between agencies, non-profit groups, growers, Indigenous and other communities in order to successfully implement conservation aquaculture. Thus, our Olympia oyster case study provides a model for engaging diverse stakeholders to recommend strategic use of conservation aquaculture where the rewards outweigh risks.

## Supporting information

**S1 Fig. Boundaries among sub-basins of the two largest estuaries on this coast.** For the Washington sub-basins, the regional team delineated these areas by integrating known oceanographic features, such as sills and straits, with established sub-basin divisions outlined by Washington Department of Fish & Wildlife and expert opinion provided by local marine scientists and restoration practitioners [81, 82]. For BC, the team delineated areas based on published work by the Department of Fisheries and Oceans in combination with expert opinion. For San Francisco Bay, California, the regional team delineated sub-basins based on extensive studies of this region [83] together with expert opinion from regional scientists. Numbers are the same as used in Figs 1 and 2 in the main paper; additional estuaries that were initially considered but were excluded from the index calculations due to insufficient data are labeled with letters.
(TIF)

**S2 Fig. Map of project type indices at the ten sites scoring highest on the ecological priority index.** Bar heights correspond to site specific values in Fig 2 in the main paper.
(TIF)

**S1 Table. Stakeholders who contributed to the development and scoring of the reward vs. risk tables and conservation aquaculture indices.**
(PDF)

**S2 Table. Scoring guidance for all criteria.**
(PDF)

**S3 Table. Detailed version of conservation aquaculture scores and indices in Excel, including scores and rationale for them for all 66 estuaries, and including formulae for calculating indices, so that users can add new sites, new data, or change weightings using this template.**
(XLSX)

## Acknowledgments

We are indebted to our stakeholder team (S1 Table) who contributed to the evaluation of risks and rewards and scoring of estuaries. We thank A. Chang and T. McClintock for specific information associated with estuary selection and the creation of sub-basins within larger complexes. In addition, we thank M. Bitter, M. Camara, J. Moore, D. Hedgecock and B. Valdopalas for help with assessing rewards and risks including population genetics issues and responses to future climate conditions. For initial development of maps and figures, we thank J. Brun, R. Saldivar and M. Klope. We acknowledge the use of imagery provided by services from NASA's Global Imagery Browse Services (GIBS), part of NASA's Earth Observing System Data and Information System (EOSDIS). This work resulted from the Science for Nature and People Partnership (SNAPP) Conservation Aquaculture Working Group. SNAPP is a partnership of The Nature Conservancy, Wildlife Conservation Society and the National Center for Ecological Analysis and Synthesis at the University of California, Santa Barbara.

## Author Contributions

**Conceptualization:** April D. Ridlon, Kerstin Wasson, Tiffany Waters, Edwin D. Grosholz.

**Formal analysis:** April D. Ridlon, Kerstin Wasson, Tiffany Waters, Edwin D. Grosholz.

**Funding acquisition:** April D. Ridlon, Kerstin Wasson, Tiffany Waters, Edwin D. Grosholz.

**Investigation:** April D. Ridlon, Kerstin Wasson, Tiffany Waters, John Adams, Jamie Donatuto, Gary Fleener, Rhona Govender, Julio Lorda, Betsy Peabody, Gifford Pinchot IV, Steven S. Rumrill, Elizabeth Tobin, Chela J. Zabin, Danielle Zacherl, Edwin D. Grosholz.

**Methodology:** April D. Ridlon, Kerstin Wasson, Tiffany Waters, John Adams, Jamie Donatuto, Gary Fleener, Rhona Govender, Julio Lorda, Betsy Peabody, Gifford Pinchot IV, Steven S. Rumrill, Elizabeth Tobin, Chela J. Zabin, Danielle Zacherl, Edwin D. Grosholz.

**Project administration:** April D. Ridlon, Kerstin Wasson.

**Resources:** Tiffany Waters, Edwin D. Grosholz.

**Software:** Aaron Kornbluth.

**Supervision:** Kerstin Wasson.

**Validation:** April D. Ridlon, Tiffany Waters, Edwin D. Grosholz.

**Visualization:** April D. Ridlon, Kerstin Wasson, Tiffany Waters, Halley Froehlich, Aaron Kornbluth.

**Writing – original draft:** April D. Ridlon, Kerstin Wasson, Tiffany Waters, Edwin D. Grosholz.

**Writing – review & editing:** April D. Ridlon, Kerstin Wasson, Tiffany Waters, John Adams, Jamie Donatuto, Gary Fleener, Halley Froehlich, Rhona Govender, Aaron Kornbluth, Julio Lorda, Betsy Peabody, Gifford Pinchot IV, Steven S. Rumrill, Elizabeth Tobin, Chela J. Zabin, Danielle Zacherl, Edwin D. Grosholz.

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
