## [Decision Letter · Decision Letter 0]

25 Jan 2021

PONE-D-20-38508

Conservation Aquaculture as a Tool for Imperiled Marine Species: Evaluation of Opportunities and Risks for Olympia Oysters

PLOS ONE

Dear Dr. Ridlon,

Thank you for submitting your manuscript to PLOS ONE. After careful consideration, we feel that it has merit but does not fully meet PLOS ONE’s publication criteria as it currently stands. Therefore, we invite you to submit a revised version of the manuscript that addresses the points raised during the review process.

I enjoyed reading this MS on restoration/conservation aquaculture. It provides an avenue for degraded fisheries and ecosystem to implement the methodology. Please, address the comments of the two reviewers. I am looking forward to reading your response and the revised MS.

We look forward to receiving your revised manuscript.

Kind regards,

Ismael Aaron Kimirei, Ph.D.

Academic Editor

PLOS ONE

Journal Requirements:

3. We note that Figure 4 includes an image of a participant in the study. 

"The authors have declared that no competing interests exist.

We note that one or more of the authors are employed by a commercial company: Hog Island Oyster Co.

4.1. Please provide an amended Funding Statement declaring this commercial affiliation, as well as a statement regarding the Role of Funders in your study. If the funding organization did not play a role in the study design, data collection and analysis, decision to publish, or preparation of the manuscript and only provided financial support in the form of authors' salaries and/or research materials, please review your statements relating to the author contributions, and ensure you have specifically and accurately indicated the role(s) that these authors had in your study. You can update author roles in the Author Contributions section of the online submission form.

4.2. Please also provide an updated Competing Interests Statement declaring this commercial affiliation along with any other relevant declarations relating to employment, consultancy, patents, products in development, or marketed products, etc.  

5. We note that Figures 1, S1 and S2 in your submission contain map/satellite images which may be copyrighted. All PLOS content is published under the Creative Commons Attribution License (CC BY 4.0), which means that the manuscript, images, and Supporting Information files will be freely available online, and any third party is permitted to access, download, copy, distribute, and use these materials in any way, even commercially, with proper attribution. For these reasons, we cannot publish previously copyrighted maps or satellite images created using proprietary data, such as Google software (Google Maps, Street View, and Earth). For more information, see our copyright guidelines: http://journals.plos.org/plosone/s/licenses-and-copyright.

5.1.    You may seek permission from the original copyright holder of Figures 1, S1 and S2 to publish the content specifically under the CC BY 4.0 license. 

5.2.    If you are unable to obtain permission from the original copyright holder to publish these figures under the CC BY 4.0 license or if the copyright holder’s requirements are incompatible with the CC BY 4.0 license, please either i) remove the figure or ii) supply a replacement figure that complies with the CC BY 4.0 license. Please check copyright information on all replacement figures and update the figure caption with source information. If applicable, please specify in the figure caption text when a figure is similar but not identical to the original image and is therefore for illustrative purposes only.

Reviewers' comments:

Reviewer's Responses to Questions

**Comments to the Author**

1. Is the manuscript technically sound, and do the data support the conclusions?

Reviewer #1: Partly

Reviewer #2: Yes

2. Has the statistical analysis been performed appropriately and rigorously? 

Reviewer #1: N/A

Reviewer #2: N/A

3. Have the authors made all data underlying the findings in their manuscript fully available?

Reviewer #1: No

Reviewer #2: Yes

4. Is the manuscript presented in an intelligible fashion and written in standard English?

Reviewer #1: No

Reviewer #2: Yes

5. Review Comments to the Author

Reviewer #1: This study titled “Conservation Aquaculture as a Tool for Imperiled Marine Species: Evaluation of Opportunities and Risks for Olympia Oyster” characterized the benefits and risks of Olympia oyster aquaculture conservation by using four criteria: social, cultural, economic and conservation. Generally, the initial idea is relatively good for conservation. However, the write up of the manuscript is too long and lacks focus. The authors presented too much information without filtering for key issues to provide a clear message to readers. The whole manuscript is boring to read and difficult to follow. The methods have no scientific background to support the issues mentioned. Specific issues on the manuscript are given below and on the attached annotated PDF.

Abstract

1. Too much information is presented on background. Reducing the text on this part will make the abstract more focused.

2. The method section is not adequately given.

3. The results of the study are inadequately presented. More results should be presented.

4. The conclusion is not clear enough to readers.

Introduction

1. The introduction is generally long, repetitive and unfocused.

2. The scientific problem the study intends to solve is not shown clearly. The authors should reduce the long introduction in order to provide a clear scientific problem.

3. The objective is also not clear enough to readers.

Methods

The methods need to be improved.

1. It long and unfocused.

2. It includes unnecessary information for a publication.

3. It uses uncommon fonts and styles.

4. It is very difficult to follow and understand what the authors did.

Results

The results should be shorted by highlighting only the important results. Can the results and discussion be presented together?

Discussion

The discussion is too long. It should be shortened. Discuss briefly the important results and not everything.

Reviewer #2: GENERAL COMMENTS

The manuscripts presents findings from a study that aimed at assessing potential risks and rewards associated with a conservation aquaculture as an important tool to support recovery of declining species particularly Olympia oysters (Ostrea lurida). The study also came up with key recommendations for establishing conservation aquaculture. Generally the manuscript is fairly well written. However, there are few areas that needs specific attention before is recommended for publication in the journal as follows:

1) Authors are advised to strongly minimize personalization of the work such as we, our study etc! Although it gives a good story but this is more of impressing than communicating. In science we write to communicate facts.

2) Authors are trying to put too many or multiple things within a single sentence which is in most cases may confuse readers. “A sentence should express a single thought or proposition”. Authors should avoid long and rambling sentences.

3) It is advisable to have section where key recommendations from this study can be found.

TITLE

It is okay. However, inclusion of a scientific name of Olympia oyster at this stage could add a lot of value.

ABSTRACT

It is okay. Authors may consider inclusion of a sentence that provides some principal findings of this study.

INTRODUCTION

Lines 39 to 42: Start with giving worldwide figure and then those from United States of America. Also, provide the estimated figure for loss of coral reefs.

MATERIALS AND METHODS

1) Lines I61 to 164: In the S1 Table, to my opinion, multiple categories could have their own specific category just like the way it is for say those who fall specifically to Grower (G), Manager or Resource Agency representative (M), NGO member (N), Conservation Scientist (S), or Tribal Representative (T). Those who fall under multiple categories should be treated separately and their opinion provided. This information should also be indicated somewhere in lines 161 to 164.

2) Line 193: Please refine the sentence to make it clearer.

3) Line 194 (for Tables 2 & 3) compared with line 224 (Table 1). Tables and figures should be cited serially by starting with 1 (Start with Table 1 or Figure 1). Thus the arrangement could be Table 2 changed to Table 1; Table 3 changed to Table 1 and finally Table 1 changed to Table 3 and cited accordingly for consistency purposes.

4) Line 202 and part of line 203: This is information plus the references can be moved to somewhere in the introductory part. In this section just tell the reader the area focused by the study.

5) Lines 215 to 216: Can Authors show the criteria used to delineate the sub-basins?

6) Lines 334 to 338 (many others): It is fortunate that authors are good in English language. Otherwise, most of the sentences are very long! Imagine a sentence of 63 words!! Not recommended for scientific writings!

RESULTS

Please see some track changes in this section in the main manuscript.

DISCUSSION

1) Line 436: Unpublished data cannot be accessible by readers. As such it is not recommended in scientific publications.

2) Lines 443 to 447: Very long sentence. Readers may not enjoy reading long sentences! Again a reference to those protocols or even a website link where those institutions can found could be very useful to readers.

3) Line 460 and several other places: The use of phrases such as “…..our stakeholders” is monotonous in this manuscript. Authors can frame this work in such a way that such phrases only remain in the Materials and Methods section. Then, make sure that whatever is presented in the results and discussed in the discussion section reflects the findings from stakeholder’s views. I believe this I believe this is doable.

4) Lines 510 to 516: How can 77 words be in a single sentence!! Readers cannot follow! There is a time where readers may lose the flow contents!

5) Lines 557 to 559: Something is missing in this sentence. Please make it clear.

CONCLUSION

The conclusion is somewhat okay. An opening sentence summarising the overall objectives of the study followed by key take home messages would add a lot of value. Authors should strictly provide the main conclusion as a result of carrying this study. Please try to reduce the “hows” here i.e the way the study was undertaken.

6. PLOS authors have the option to publish the peer review history of their article (what does this mean?). If published, this will include your full peer review and any attached files.

Reviewer #1: No

Reviewer #2: **Yes: **Amon Paul Shoko

---

## [Author Response · Author response to Decision Letter 0]

23 Mar 2021

Review Comments to the Author

Reviewer #1 (Anonymous): This study titled “Conservation Aquaculture as a Tool for Imperiled Marine Species: Evaluation of Opportunities and Risks for Olympia Oyster” characterized the benefits and risks of Olympia oyster aquaculture conservation by using four criteria: social, cultural, economic and conservation. Generally, the initial idea is relatively good for conservation. However, the write up of the manuscript is too long and lacks focus. The authors presented too much information without filtering for key issues to provide a clear message to readers. The whole manuscript is boring to read and difficult to follow. The methods have no scientific background to support the issues mentioned. Specific issues on the manuscript are given below and on the attached annotated PDF.

We are pleased that the reviewer sees the conservation value that this manuscript provides. We appreciated and incorporated a great many of the reviewer’s suggestions to cut back on the overall length of the manuscript, particularly in the introduction, and to sharpen the focus of the text in some sections, which we address with the specific comments below. We also provided additional subheadings in the Discussion to increase structure and ease of reading. 

Abstract

We thank the reviewer for these guiding comments about the Abstract and have completely rewritten this section. In each section, we indicate general changes made. 

1. Too much information is presented on background. Reducing the text on this part will make the abstract more focused.

We have reduced the background information.

2. The method section is not adequately given.

We believe that the abstract now better highlights both the engaging of a collaborative team and the quantitative methods we used to evaluate locations and projects. 

3. The results of the study are inadequately presented. More results should be presented.

We re-wrote the abstract to include results specific to Olympia oysters, including key risks and rewards of using aquaculture as a tool for this species, top ecological priority areas for expansion and their shared trends (i.e. recruitment limitation and isolation), and the applicability of different project types across estuaries. 

4. The conclusion is not clear enough to readers

We edited the last 2 sentences of the abstract to clarify the conclusion.

Introduction

1. The introduction is generally long, repetitive and unfocused.

We thank the reviewer for the specific suggestions provided in the annotated pdf. Using these as a guide, we have refocused the introduction, cutting out repetitive text, rearranging information and greatly reducing the overall length. 

2. The scientific problem the study intends to solve is not shown clearly. The authors should reduce the long introduction in order to provide a clear scientific problem.

We have shortened and refocused the introduction to address this comment, and thank the reviewer for their specific in-line edits, which guided our revision of this section. Our scientific problem is stated as such: “To our knowledge, there has been no thorough evaluation of the rewards versus risks of conservation aquaculture to guide strategic planning for any species on this coast, let alone for Olympia oysters. Before funders, planners, regulators, conservation organizations, or growers move forward with aquaculture initiatives, there is a pressing need to conduct a robust, collaborative evaluation of whether and under what conditions the rewards outweigh the risks. Additionally, detailed spatial analyses are needed to identify the specific locations where conservation aquaculture efforts can be expected to have a high likelihood for success.”

3. The objective is also not clear enough to readers.

We specify our goal/objective in the introduction: 

“Our goal was to conduct strategic planning and develop decision-support tools to identify the use of aquaculture to support recovery of Olympia oysters across the range of the species.”

Methods

The methods need to be improved.

1. It long and unfocused.

We incorporated some of the reviewer’s specific edits provided in the annotated pdf to ensure more concise, focused writing in this section.

2. It includes unnecessary information for a publication.

Unfortunately, we cannot tell from this comment which information the reviewer feels is unnecessary for a publication. Again, where the reviewer made specific edits that shortened and focused the text in this section, we incorporated them.

3. It uses uncommon fonts and styles.

Thank you for catching these font inconsistencies. We have corrected them in the revised text.

4. It is very difficult to follow and understand what the authors did.

We incorporated any edits the reviewer provided that shortened and/or focused the text in this section in order to make the methods easier for the reader to follow. 

Results

The results should be shorted by highlighting only the important results. Can the results and discussion be presented together?

The idea of combining the results and discussion is one that was raised among our co-authors as well. We also believe that these types of results are best presented in a mixed results and discussion format, but this violates the journal format, unfortunately. To address this, we kept the results as brief as possible and the discussion is lengthier than it would otherwise be.

Discussion

The discussion is too long. It should be shortened. Discuss briefly the important results and not everything.

Please see the previous comment about the length of the discussion. We provided additional subheadings to the discussion to further focus it, and accepted any specific edits provided in the annotated .pdf that shortened and/or focused the text in this section.

Reviewer #2, Dr. Amon Paul Shoko 

GENERAL COMMENTS The manuscripts presents findings from a study that aimed at assessing potential risks and rewards associated with a conservation aquaculture as an important tool to support recovery of declining species particularly Olympia oysters (Ostrea lurida). The study also came up with key recommendations for establishing conservation aquaculture. Generally the manuscript is fairly well written. 

We thank Dr. Shoko for his detailed review. The revisions that he has suggested have certainly improved the clarity of our writing. We believe that these revisions make the paper easier to read, and thus our messages easier to understand. We detail the ways in which we’ve incorporated his suggestions below. 

However, there are few areas that needs specific attention before is recommended for publication in the journal as follows:

1) Authors are advised to strongly minimize personalization of the work such as we, our study etc! Although it gives a good story but this is more of impressing than communicating. In science we write to communicate facts.

Some of the terms that connote personalization (“stakeholders” “we”, etc.) have been changed to address this style comment, especially in places in the text where the reviewer suggests this (in the annotated pdf provided). However, we continue to use the active voice, particularly in the Methods section, to accurately describe our process.

2) Authors are trying to put too many or multiple things within a single sentence which is in most cases may confuse readers. “A sentence should express a single thought or proposition”. Authors should avoid long and rambling sentences.

We thank you for highlighting sentences that were too long, and for pointing out that audiences for whom English is a second language might be especially confused by this type of writing. We appreciate the opportunity to make our writing clearer and more accessible to all audiences, and have taken the reviewer’s suggestions to break up long sentences in every instance. 

3) It is advisable to have section where key recommendations from this study can be found.

We provide key recommendations for conservation aquaculture with Olympia oysters specifically by beneficiary/user group, and then by location/region in the Discussion. We have added a number of subheadings to clarify the location of recommendations in the Discussion (e.g. “Regional Recommendations”). With Figure 3, we provide an explicit process which we recommend to create similar strategic planning for the use of aquaculture to support other imperiled species. 

TITLE

It is okay. However, inclusion of a scientific name of Olympia oyster at this stage could add a lot of value.

We have added the scientific name to the title

ABSTRACT

It is okay. Authors may consider inclusion of a sentence that provides some principal findings of this study.

We thank the reviewer for this comment and have completely rewritten this section to include some principle findings specific to Olympia oysters, including key risks and rewards of aquaculture as a tool for this species, top ecological priority areas for expansion and their shared trends (i.e. recruitment limitation and isolation), and the applicability of different project types across estuaries. 

INTRODUCTION

Lines 39 to 42: Start with giving worldwide figure and then those from United States of America. Also, provide the estimated figure for loss of coral reefs.

While we appreciate this suggestion, the figure of oyster loss from the U.S. is the most relevant to the species we are evaluating, and changing the order of these statistics will obscure that. The citation provided will give global context for oyster loss, and the comparison to coral reef losses in more detail for interested readers.

MATERIALS AND METHODS

1) Lines I61 to 164: In the S1 Table, to my opinion, multiple categories could have their own specific category just like the way it is for say those who fall specifically to Grower (G), Manager or Resource Agency representative (M), NGO member (N), Conservation Scientist (S), or Tribal Representative (T). Those who fall under multiple categories should be treated separately and their opinion provided. This information should also be indicated somewhere in lines 161 to 164.

We thank the reviewer for his suggestion. However, we feel that having an additional “many categories” category would obscure the point of this table, which is to make stakeholder identity clear. We generated these categories to differentiate stakeholders from one another (e.g. we use “conservation scientists” to describe academic PIs, but this term could also be used to describe researchers employed by NGOs), and asked each person to self-identify the role they would be assuming while working on this project. To clarify this point, we have added the following line to the legend for S1 Table: “While some fall into multiple categories, we asked each stakeholder to identify the primary role that they assumed to contribute to this project.”

2) Line 193: Please refine the sentence to make it clearer.

We edited this sentence for clarity. 

3) Line 194 (for Tables 2 & 3) compared with line 224 (Table 1). Tables and figures should be cited serially by starting with 1 (Start with Table 1 or Figure 1). Thus the arrangement could be Table 2 changed to Table 1; Table 3 changed to Table 1 and finally Table 1 changed to Table 3 and cited accordingly for consistency purposes.

Thank you for pointing out this inconsistency in the text. We have removed the reference to Tables 2 and 3 in the Methods section so that the tables are cited serially in the order that they are numbered. 

4) Line 202 and part of line 203: This is information plus the references can be moved to somewhere in the introductory part. In this section just tell the reader the area focused by the study.

We agree that normally a methods section doesn’t include citations or background. However, this sentence is meant to context the methods immediately following, as it speaks to why we focused on estuaries vs. the open coast. We also suspect that such a minor detail would be lost in an already long introduction. For these reasons, we feel this is well placed. 

5) Lines 215 to 216: Can Authors show the criteria used to delineate the sub-basins?

We agree that we need to elaborate on these criteria. They are now outlined in the legend for S1 Fig, and three relevant citations have been added to the references:

 “ Boundaries among sub-basins of the two largest estuaries on this coast. For the Washington sub-basins, the regional team delineated these areas by integrating known oceanographic features, such as sills and straits, with established sub-basin divisions outlined by Washington Department of Fish & Wildlife and expert opinion provided by local marine scientists and restoration practitioners [81, 82]. For BC, the team delineated areas based on published work by the Department of Fisheries and Oceans in combination with expert opinion. For San Francisco Bay, California, the regional team delineated sub-basins based on extensive studies of this region [83] together with expert opinion from regional scientists. Numbers are the same as used in Figs. 1-2 in the main paper; additional estuaries that were initially considered but were excluded from the index calculations due to insufficient data are labeled with letters.” 

6) Lines 334 to 338 (many others): It is fortunate that authors are good in English language. Otherwise, most of the sentences are very long! Imagine a sentence of 63 words!! Not recommended for scientific writings!

We thank the reviewer for bringing this long sentence to our attention. It has been broken up into 2 sentences, both edited for clarity.

RESULTS

Please see some track changes in this section in the main manuscript.

We thank the reviewer for his comments in the Results section.

DISCUSSION

1) Line 436: Unpublished data cannot be accessible by readers. As such it is not recommended in scientific publications.

We have removed the reference to unpublished data.

2) Lines 443 to 447: Very long sentence. Readers may not enjoy reading long sentences! Again a reference to those protocols or even a website link where those institutions can found could be very useful to readers.

We thank the reviewer for pointing this long sentence out. We have broken the sentence into 2, eliminated the long hatchery name and used an existing citation to reference the hatchery protocols. 

3) Line 460 and several other places: The use of phrases such as “…..our stakeholders” is monotonous in this manuscript. Authors can frame this work in such a way that such phrases only remain in the Materials and Methods section. Then, make sure that whatever is presented in the results and discussed in the discussion section reflects the findings from stakeholder’s views. I believe this I believe this is doable.

Some of the terms that connote personalization (“stakeholders” “we”, etc.) have been changed to address this style comment, especially in places in the text where the reviewer suggested this (in the annotated pdf provided). We especially took note of places in which these phrases were monotonous or overused and adjusted the text. Please see our general comment above about why we did not eliminate all of this language.

4) Lines 510 to 516: How can 77 words be in a single sentence!! Readers cannot follow! There is a time where readers may lose the flow contents!

We thank the reviewer for pointing this long sentence out. We have broken the sentence into 3 separate sentences for clarity.

5) Lines 557 to 559: Something is missing in this sentence. Please make it clear.

We have edited this line to read:

“We also recommend that conservation aquaculture projects that involve harvest include a component of public education about not harvesting or disturbing wild Olympia oyster populations that are on restoration sites or in non-harvestable areas.”

CONCLUSION

The conclusion is somewhat okay. An opening sentence summarising the overall objectives of the study followed by key take home messages would add a lot of value. Authors should strictly provide the main conclusion as a result of carrying this study. Please try to reduce the “hows” here i.e the way the study was undertaken.

The opening sentence concluding the importance of the work is: 

“The new approach and assessment tools developed to evaluate conservation aquaculture for Olympia oysters provide a template that is clearly needed to evaluate the potential use of conservation aquaculture for other species that have undergone declines, and especially in evaluating the tradeoffs between rewards and risks where aquaculture is already being used as a tool to address declines (e.g. reef-building corals).” 

We feel additional take-home messages in this section would lengthen an already very long discussion. Any reference to how the study was undertaken (using a collaborative process with diverse stakeholders) is a conclusion/main take away message, because the process we used to evaluate this tool is a product of the study as much as the index framework we created.

---

## [Decision Letter · Decision Letter 1]

24 May 2021

Conservation Aquaculture as a Tool for Imperiled Marine Species: Evaluation of Opportunities and Risks for Olympia Oysters, * Ostrea lurida*

*PONE-D-20-38508R1*

*Dear Dr. Ridlon,*

*We’re pleased to inform you that your manuscript has been judged scientifically suitable for publication and will be formally accepted for publication once it meets all outstanding technical requirements.*

*Within one week, you’ll receive an e-mail detailing the required amendments. When these have been addressed, you’ll receive a formal acceptance letter and your manuscript will be scheduled for publication.*

*An invoice for payment will follow shortly after the formal acceptance. To ensure an efficient process, please log into Editorial Manager at http://www.editorialmanager.com/pone/, click the 'Update My Information' link at the top of the page, and double check that your user information is up-to-date. If you have any billing related questions, please contact our Author Billing department directly at authorbilling@plos.org.*

*If your institution or institutions have a press office, please notify them about your upcoming paper to help maximize its impact. If they’ll be preparing press materials, please inform our press team as soon as possible -- no later than 48 hours after receiving the formal acceptance. Your manuscript will remain under strict press embargo until 2 pm Eastern Time on the date of publication. For more information, please contact onepress@plos.org.*

*Kind regards,*

*Ismael Aaron Kimirei, Ph.D.*

*Academic Editor*

*PLOS ONE*

* *

*Additional Editor Comments (optional):*

Please respond to the minor comments made by reviewer #2 

* *

*Reviewers' comments:*

*Reviewer's Responses to Questions*

***Comments to the Author***

*1. If the authors have adequately addressed your comments raised in a previous round of review and you feel that this manuscript is now acceptable for publication, you may indicate that here to bypass the “Comments to the Author” section, enter your conflict of interest statement in the “Confidential to Editor” section, and submit your "Accept" recommendation.*

*Reviewer #1: All comments have been addressed*

*Reviewer #2: (No Response)*

*2. Is the manuscript technically sound, and do the data support the conclusions?*

*The manuscript must describe a technically sound piece of scientific research with data that supports the conclusions. Experiments must have been conducted rigorously, with appropriate controls, replication, and sample sizes. The conclusions must be drawn appropriately based on the data presented. *

*Reviewer #1: Yes*

*Reviewer #2: Yes*

*3. Has the statistical analysis been performed appropriately and rigorously? *

*Reviewer #1: N/A*

*Reviewer #2: No*

*4. Have the authors made all data underlying the findings in their manuscript fully available?*

*The PLOS Data policy requires authors to make all data underlying the findings described in their manuscript fully available without restriction, with rare exception (please refer to the Data Availability Statement in the manuscript PDF file). The data should be provided as part of the manuscript or its supporting information, or deposited to a public repository. For example, in addition to summary statistics, the data points behind means, medians and variance measures should be available. If there are restrictions on publicly sharing data—e.g. participant privacy or use of data from a third party—those must be specified.*

*Reviewer #1: Yes*

*Reviewer #2: Yes*

*5. Is the manuscript presented in an intelligible fashion and written in standard English?*

*PLOS ONE does not copyedit accepted manuscripts, so the language in submitted articles must be clear, correct, and unambiguous. Any typographical or grammatical errors should be corrected at revision, so please note any specific errors here.*

*Reviewer #1: Yes*

*Reviewer #2: (No Response)*

*6. Review Comments to the Author*

*Please use the space provided to explain your answers to the questions above. You may also include additional comments for the author, including concerns about dual publication, research ethics, or publication ethics. (Please upload your review as an attachment if it exceeds 20,000 characters)*

*Reviewer #1: The said subtitles on the discussion section in the response were present in the original file. Very little improvement has been made on reducing the length of the manuscript.*

*Reviewer #2: 3. Has the statistical analysis been performed appropriately and rigorously?*

*Actually this is the area that needs some clarification. Is there any statistical analysis undertaken in this work or was it not necessary? How can the results be justifiable scientifically? Is there any good explanation to this observation!*

*The authors have tried to address most of the comments. However, clarification is needed on statistical analysis as indicated in question 3 above. Do the authors mean that the nature of data did not allow for any kind of statistical analysis!?*

*7. PLOS authors have the option to publish the peer review history of their article (what does this mean?). If published, this will include your full peer review and any attached files.*

**

**

*Reviewer #1: No*

*Reviewer #2: No*

---

## [Editor Report · Acceptance letter]

10 Jun 2021

PONE-D-20-38508R1 

Conservation Aquaculture as a Tool for Imperiled Marine Species: Evaluation of Opportunities and Risks for Olympia Oysters, *Ostrea lurida*

Dear Dr. Ridlon:

I'm pleased to inform you that your manuscript has been deemed suitable for publication in PLOS ONE. Congratulations! Your manuscript is now with our production department. 

Kind regards, 

on behalf of

Dr. Ismael Aaron Kimirei 

Academic Editor

PLOS ONE